# Dynamic Neural Fortresses: An Adaptive Shield for Model Extraction Defense

**Siyu Luan**[1]*, **Zhenyi Wang**[2]#*, **Li Shen**[3], **Zonghua Gu**[4], **Chao Wu**[5], **Dacheng Tao**[6]
[1]University of Copenhagen, Denmark [2]University of Maryland, College Park, USA
[3] Shenzhen Campus of Sun Yat-sen University [4]Hofstra University, USA
[5]University at Buffalo, USA [6]Nanyang Technological University, Singapore
`siyu.luan.sund@gmail.com, zwang169@umd.edu`

## Abstract

Model extraction aims to acquire a pre-trained black-box model concealed behind a black-box API. Existing defense strategies against model extraction primarily concentrate on preventing the unauthorized extraction of API functionality. However, two significant challenges still need to be solved: (i) Neural network architecture of the API constitutes a form of intellectual property that also requires protection; (ii) The current practice of allocating the same network architecture to both attack and benign queries results in substantial resource wastage. To address these challenges, we propose a novel *Dynamic Neural Fortresses* (DNF) defense method, employing a dynamic Early-Exit neural network, deviating from the conventional fixed architecture. Firstly, we facilitate the random exit of attack queries from the network at earlier layers. This strategic exit point selection significantly reduces the computational cost for attack queries. Furthermore, the random exit of attack queries from earlier layers introduces increased uncertainty for attackers attempting to discern the exact architecture, thereby enhancing architectural protection. On the contrary, we aim to facilitate benign queries to exit at later layers, preserving model utility, as these layers typically yield meaningful information. Extensive experiments on defending against various model extraction scenarios and datasets demonstrate the effectiveness of DNF, achieving a notable $2\times$ improvement in efficiency and an impressive reduction of up to 12% in clone model accuracy compared to SOTA defense methods. Additionally, DNF provides strong protection against neural architecture theft, effectively safeguarding network architecture from being stolen.

## 1 Introduction

Deep learning models have found extensive application in diverse real-world scenarios. Major tech companies such as OpenAI, Google, Meta, and Amazon have developed a range of black-box APIs, offering services to users. This service-oriented paradigm is commonly referred to as Machine Learning as a Service (MLaaS) (Ribeiro et al., 2015). The black-box pre-trained models provided by these entities hold significant business value. However, model extraction (Oliynyk et al., 2023) aims to steal the functionality of the victim model with some query data only. Consequently, safeguarding these APIs from model extraction has become an urgent and critical task.

Existing model extraction defense methods primarily concentrate on safeguarding the functionality of the API from stealing. However, two significant challenges still need to be solved. Firstly, network architecture itself can be considered intellectual property (IP) and requires protection against theft. This is particularly pertinent as determining the optimal architecture often involves substantial computational resources and extensive hyperparameter search (Wistuba et al., 2019). In addition, neural architecture stealing (Oh et al., 2018; Rolnick & Kording, 2020; Zhu et al., 2021; Carlini et al., 2024) is attracting growing interest, driven by the significant commercial value of proprietary model architectures used in production systems. Secondly, current defense strategies allocate the same computational power to both benign and attack queries. We contend that this approach is inappropriate, given that attack queries do not demand extensive computational resources.

---

*Siyu Luan and Zhenyi Wang contribute equally. Corresponding author: Zhenyi Wang

To overcome these limitations, we propose a novel defense approach termed *Dynamic Neural Fortresses* (DNF). This method utilizes a dynamic Early-Exit neural network (EENN) to enhance the efficacy of defenses against model extraction. Our objective is to facilitate the *random* exit of attack queries at the early initial layers of the network, as these layers typically yield non-semantic and misleading information for attackers. This strategy minimizes the consumption of computational resources for attack queries, leading to a significant improvement in overall running efficiency. Furthermore, by enabling attack queries to randomly exit at different layers of the victim model, it amplifies the challenge of accurately discerning the exact network architecture of the victim model for attackers. The substantial uncertainty introduced regarding the model architecture serves to safeguard the intellectual property of the model. On the other hand, we aim to enable benign queries to exit at later layers of the network, as these layers typically output meaningful and semantic information, making them suitable for providing valuable insights to legitimate users.

To implement our strategy, we draw inspiration from the deep information bottleneck theory (Alemi et al., 2017). Specifically, when dealing with out-of-distribution (OOD) data, we undertake a two-fold approach. Firstly, we minimize the mutual information between latent features and data labels to deliberately reduce the model's prediction ability, ensuring effective defense. Concurrently, we maximize the mutual information between input data and latent features. This action allows the feature extractor to capture non-semantic information in the input data, leading to deliberately inaccurate model predictions for enhanced defense. Additionally, this learning objective leads to non-deterministic exits in the classifier to fortify the architecture against potential theft. Conversely, when handling in-distribution (ID) data, our strategy adjusts to maintain model utility. We maximize the mutual information between latent features and data labels, thereby enhancing the association between these features and data labels to improve overall model performance on benign queries. Simultaneously, we minimize the mutual information between input data and latent features. This step enables the feature extractor to capture both compressive and semantic information present in the input data, contributing to a more utility-driven model. We illustrate our approach in Figure 1.

To assess the efficacy of the proposed method, we conduct comprehensive experiments aimed at defending against both data-based model extraction (where the attacker uses similar real data to query the victim model) and data-free model extraction (where the attacker uses synthetic data only to query the victim model) in both soft-label and hard-label attack settings. Our method is highly adaptable and can be applied seamlessly to various victim model architectures, including both ResNet and large-scale pre-trained Vision Transformer (ViT) models. Its simple design and ease of implementation, using standard linear classifiers without requiring specialized exit classifiers, highlight its exceptional scalability and flexibility. The results demonstrate that our approach consistently surpasses the state-of-the-art (SOTA) defense method, achieving up to a 12% reduction in clone model accuracy. Simultaneously, our method significantly enhances running efficiency compared to SOTA defense methods, achieving $2\times$ speedup. Additionally, our method outperforms other defense techniques in terms of overall model utility. Crucially, our defense is notably more effective in model extraction scenarios, regardless of whether attackers utilize OOD data or have access to in-distribution data, highlighting its broad applicability. Furthermore, we conducted additional experiments to evaluate our defense against the SOTA model architecture stealing method (Carlini et al., 2024). The results demonstrate that our approach can effectively protect model architecture from theft.

Our main contributions are summarized as three-fold:

- We present the first defense framework that provides three key protective benefits simultaneously: (1) safeguarding the model's functionality while substantially lowering the accuracy of cloned models, (2) securing the model architecture, and (3) greatly improving efficiency in defending against various model extraction scenarios.

- We propose a novel dynamic Early-Exit network defense method with adaptive deep variational information bottleneck learning objective. To the best of our knowledge, this is the first attempt to introduce Early-Exit network into the field of model extraction defense.

- Extensive experiments on defending against various model extraction attack settings demonstrate the superiority of the proposed method compared to SOTA defense methods.

## 2 RELATED WORK

**Model Extraction Attack** (a) *Functionality Stealing*: One of the objectives of model extraction is to achieve the same functionality as the victim model or to make their predictions consistent with those of the victim model (Oliynyk et al., 2023; Nayak et al., 2019; Pal et al., 2020; Jagielski et al., 2020; Li et al., 2023). According to the data type used for model extraction, the techniques used in model extraction can be further classified into: (1) *Data-based Model Extraction (DBME)*: DBME is a technique that assumes attackers either have knowledge of the training dataset used for the target victim model or possess a surrogate dataset to extract knowledge from the victim model (Papernot et al., 2017; Orekondy et al., 2019; Tang et al., 2024). (2) *Data-free Model Extraction (DFME)*: DFME is a technique that assumes attackers have no prior knowledge of the training dataset and iteratively refine the dataset used to extract knowledge from the victim model based on the model's output information (Kariyappa et al., 2021a; Truong et al., 2021; Sanyal et al., 2022a; Hu et al., 2023).

(b) *Architecture Stealing*: Architecture stealing is a critical model extraction attack that seeks to uncover the internal architecture of a deployed model (Oh et al., 2018; Rolnick & Kording, 2020; Zhu et al., 2021; Carlini et al., 2024). This type of attack is gaining increasing attention due to the high commercial value of proprietary model architectures in production. Attackers target these architectures to gain valuable insights that could be used to replicate or enhance their own models.

**Model Extraction Defense** Existing model extraction defense can be categorized into two classes.

[1] *Model extraction prevention defense*. Existing methods consist of: (1) output perturbation (Orekondy et al., 2020; Mazeika et al., 2022); (2) model ensemble (Kariyappa et al., 2021b); (3) defensive training (Wang et al., 2023; Hong et al., 2024); (4) active watermarking (Wang et al., 2024). However, these methods incur significantly increased computation or memory costs due to the need for backpropagation of the victim model during deployment or the storage of multiple models.

[2] *Model extraction verification/detection defense* (Adi et al., 2018; Jia et al., 2021; Szyller et al., 2021; Dziedzic et al., 2022). There are two classes of defense methods: (1) *Detection-based defense* (Juuti et al., 2019; Kariyappa & Qureshi, 2020; Pal et al., 2021) aims to detect whether a query is benign or malicious. These defenses become ineffective when attackers change their query data distribution. Furthermore, OOD detectors are vulnerable to attacks (Azizmalayeri et al., 2022), making it easy for adversaries to bypass OOD detection-based defenses. (2) *Verification-based defense* (Adi et al., 2018; Jia et al., 2021; Szyller et al., 2021) can only confirm if a model has been stolen; they do not inherently prevent theft and *cannot reduce clone model accuracy*. Moreover, when attackers do not make the cloned model public, such methods are incapable of proving model theft.

Our approach falls under the category of *model extraction prevention defense* since our approach can maximally reduce the clone model accuracy. Existing defense strategies primarily focus on protecting the API functionality from theft. However, two significant challenges still need to be solved. First, the architecture itself holds significant intellectual property value, given the substantial resources invested in searching for the optimal network structure. Therefore, safeguarding the architecture becomes imperative. Second, the current defense methods employ the same architecture to serve both attackers and benign users, resulting in unnecessary resource wastage. In contrast, we introduce the first defense framework that offers three primary protective advantages simultaneously: (1) protecting the model's functionality while considerably decreasing the accuracy of cloned models, (2) shielding the internal model architecture, and (3) significantly boosting defense efficiency against a wide range of model extraction threats.

**Early-Exit Neural Networks** "Early Exiting" (EE) strategies (Teerapittayanon et al., 2016) incorporate exits into the early layers of DNN to achieve more efficient inference (Matsubara et al., 2022). To the best of our knowledge, this paper represents the first attempt to introduce Early-Exit neural networks (EENN) into model extraction defense. Although EENN can enhance the efficiency of standard DNNs, they are unsuitable for direct application in defending against model extraction since they treat ID and OOD data uniformly, making it challenging to simultaneously maintain model utility and achieve effective defense. In contrast, our approach involves different learning objectives tailored for ID and OOD data. This approach plays a pivotal role in preserving model utility while concurrently ensuring effective defense.

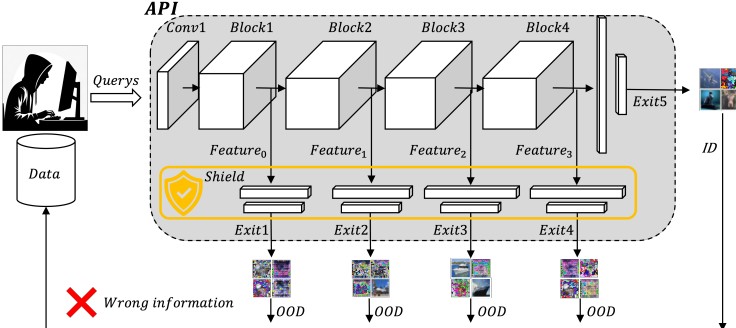

Figure 1: Illustration of DNF: ID data is expected to exit at the later layers of the EENN, after more feature extraction operations. Conversely, OOD data (attack queries) is anticipated to exit non-deterministically at earlier layers, undergoing fewer feature extraction operations. This figure shows one scenario that ID data exits from the last layer, while OOD data exits from earlier layers (It is important to note that earlier layers refer to the *relatively early exits* among all the exits of the network, while the later layers refer to the *relatively late exits* among all the exits). This learning goal serves two purposes: (1) enhancing uncertainty regarding the model architecture, preventing attackers from precisely determining the victim model's structure; (2) earlier layers may not yield meaningful features, leading the victim model to make incorrect predictions, thereby providing attackers with inaccurate information. Thus, we preserve model utility while achieving effective defense.

## 3 PRELIMINARY

**Attacker's Knowledge and Goal** During model extraction, attacker submits query samples $X = \{\boldsymbol{x}_i\}_{i=0}^{i=Q}$ to the victim model $V(\boldsymbol{x}, \boldsymbol{\delta}_V)$ parameterized by $\boldsymbol{\delta}_V$. Through $V$'s API, the attacker obtains the outputs $\boldsymbol{y}_i = V(\boldsymbol{x}_i, \boldsymbol{\delta}_V)$. Subsequently, the attacker constructs the dataset $\{(\boldsymbol{x}_i, \boldsymbol{y}_i)\}_{i=0}^{i=Q}$. The primary objective of the attacker is to employ the dataset $\{(\boldsymbol{x}_i, \boldsymbol{y}_i)\}_{i=0}^{i=Q}$ to train a clone model $C$ parameterized by $\boldsymbol{\delta}_C$, aiming to endow $C$ with similar functionality as the victim model $V$. The optimization objective for learning the clone model is: $\min_{\boldsymbol{\delta}_C} \sum_{i=0}^{i=Q} l(\boldsymbol{x}_i, \boldsymbol{y}_i, \boldsymbol{\delta}_C)$.

When an attacker possesses information about the training data used by the victim model, it is termed as *data-based model extraction* (DBME). Conversely, attacks operating without any prior knowledge of the victim model's training data are named as *data-free model extraction* (DFME). Furthermore, two distinct settings of model extraction are identified based on the API return values: *soft-label* and *hard-label* settings, depending on whether the API returns probability values or only the top-1 label. More details about model extraction can be found in Appendix A.

**Defender's Knowledge and Goal** Defender aims to minimize the test accuracy of the *clone model* on the ID test dataset while preserving the test accuracy of the *victim model* on the same ID test dataset. The defender operates with limited knowledge, unaware of the attacker's strategy, whether a query is malicious or benign, and details about the attacker's model architecture. Assumptions are widely made that the attack query data are OOD in existing model extraction attack and defense works Orekondy et al. (2019); Kariyappa & Qureshi (2020); Kariyappa et al. (2021b); Pal et al. (2020); Truong et al. (2021); Wang et al. (2023) for several reasons. Firstly, APIs typically offer users limited information, often granting access solely to input-output pairs. This lack of transparency impedes attackers from gaining insights into the specific in-distribution (ID) data used for training Wang (2021). Secondly, the in-distribution data is typically treated as a private asset by the model owner, and it is not disclosed to users due to concerns related to privacy and security. However, it's important to clarify that our assumption regarding the OOD attack query data distribution is made for presentation simplicity. Our defense method is versatile and can be applied to defend against model extraction attacks where the attacker has access to in-distribution data to query the victim model, such as in the JBDA attack (Papernot et al., 2017; Juuti et al., 2019).

## 4 METHODOLOGY

In this section, we first present the *Dynamic Neural Fortresses* (DNF) (see Figure 1) learning objective in Section 4.1. Then, we discuss the training and deployment process of DNF in Section 4.2.

## 4.1 LEARNING OBJECTIVE OF DNF

The training dataset $D_{tr}$ utilized to train the victim model is commonly referred to as the ID dataset. The parameters of the victim model, denoted as $\boldsymbol{\delta}_V$, are typically determined by optimizing:

$$\min_{\boldsymbol{\delta}_V} \left[ \mathcal{L}_0 = \mathbb{E}_{(\boldsymbol{x},y) \sim D_{tr}} \left[ L(\boldsymbol{x}, y, \boldsymbol{\delta}_V) \right] \right] \tag{1}$$

where $L(\boldsymbol{x}, y, \boldsymbol{\delta}_V)$ is the cross-entropy loss function for classification task. Traditional victim model utilizes a fixed architecture, to serve both attack and benign queries. This enables the attacker to easily steal the architecture and wastes a lot of computation resources on attack queries. To solve these issues, we propose a dynamic network solution. In the following, we first give an overview of our DNF defense method. Then, we discuss the learning objectives for ID and OOD data, respectively.

**Overview** Given a training dataset $\mathcal{D}_{tr}$ which contains $M$ classes, the initial step involves training a network, as depicted in Eq. (1). Subsequently, the pre-trained parameters $\boldsymbol{\delta}_V$ are frozen, and exit classifiers $\{V_i^*\}_{i=0}^{i=N}$ are added on top of it. $V_i^*$ denotes the $i^{th}$ exit classifier. Following this, a pre-trained OOD dataset generator $G_{ood}$ is employed to generate the OOD dataset $\mathcal{D}_{ood}$ along with corresponding labels obtained through the first exit of Early-Exit neural network inference process in $V^*$, Appendix Table 20 provides a summary of the symbol notations used throughout the paper for clarity and reference.

Given a pre-trained model $V$ parameterized by $\boldsymbol{\delta}_V$, to convert it into an Early-Exit neural network $V^*$ (EENN), as shown in Appendix Figure 4. We first choose the confidence threshold for each intermediate exit classifier during the $V^*$ inference, and intermediate exit classifiers, the number of Early-Exit $N$ (intermediate exit classifiers), and their placements over $V$. Then, we can freeze the backbone parameters of $V$, i.e., $\boldsymbol{\delta}_V$, and train the intermediate classifiers to optimize the exit classifiers $\{V_i^*\}_{i=0}^{i=N}$. Each exit classifier $V_i^*$ contains one or two Linear layers for classification, and place the classifier in the appropriate location depending on the structure of the victim model $V$, we put more details in Appendix E. We only add a tiny number of trainable parameters (account for $0.47\%$ of the victim model parameters). It is important to highlight that the exit classifiers use standard architectures without requiring any specialized design, and they can be easily integrated into the victim model in a plug-and-play fashion. This straightforward implementation works seamlessly across various architectures, demonstrating strong adaptability and flexibility.

To obfuscate information gathered by potential attackers, our goal is to maximize the disparity between exit layers for OOD and ID data. We incorporate the entropy value of each exit layer as an additional regularization measure specific to that exit layer. We denote the victim model outputs at the $i^{th}$ exit classifier as $\boldsymbol{h} = V_i^*(\boldsymbol{x}, \boldsymbol{\delta}_V^i)$. $V_{i,j}^*(\boldsymbol{x}, \boldsymbol{\delta}_V^i)$ denotes the $j^{th}$ class probability at $i^{th}$ exit classifier. $\boldsymbol{\delta}_V^i$ denotes the parameters associated with the $i^{th}$ classifier, $V_i^*$. The entropy on the victim model outputs is defined as the following (where $J$ denotes the number of classes.):

$$H(\boldsymbol{h}) = -\sum_{j=1}^{j=J} V_{i,j}^*(\boldsymbol{x}, \boldsymbol{\delta}_V^i) \log V_{i,j}^*(\boldsymbol{x}, \boldsymbol{\delta}_V^i) \tag{2}$$

The entropy value increases with the model's uncertainty at a given exit layer. Accordingly, we anticipate higher entropy for ID data at earlier exits 1 through $(n-1)$ and a lower entropy value at the later exit. This arrangement ensures that ID benign queries exit at later layers of the victim model to preserve model utility. Conversely, OOD data follows the opposite pattern, with lower entropy values at earlier exits 1 through $(n-1)$ and a higher entropy value at the later exit. This ensures that OOD attack queries exit at earlier layers, contributing to effective defense (see Figure 1). In the following, we will describe the separate learning objectives for ID and OOD data, respectively.

**ID Data Learning Objective** We present the following learning objectives Eq. (3) for ID queries.

$$\mathcal{L}_{id} = \sum_{i=1}^{i=N} [I(X_{id}; Z_i) - I(Z_i; Y_{id}) + \alpha_i K(i) H(V_i^*(\boldsymbol{x}_{id}))] \tag{3}$$

In Eq. (3), $N$ is the number of the exit classifiers, two distinct loss terms are incorporated: entropy regularization and Information Bottleneck (IB) regularization. Subsequently, we elaborate on each of these loss terms in the following.

(1) *Entropy Regularization*: $H(V_i^*(\boldsymbol{x}_{id}))$ is the entropy regularization defined in Eq. (2), $\alpha_i$ is the hyperparameter for the $i^{th}$ exit, and the piecewise function $K(i)$ assigns values based on the

comparison of the integer $i$ with the constant $N$. $K(i)$ returns a value of $1$ when $i$ is equal to the $N$. If $i$ is less than the $N$, $K(i)$ instead returns a value of $-1$. The objective is to minimize the entropy at the later layer, facilitating the ID query exit at that point, as a lower entropy value indicates a concentration of a probability distribution on a particular class, as shown in Appendix Figure 5. Simultaneously, we aim to maximize entropy at the other layers, ensuring that probabilities in those layers are more uniform and flat. This strategy discourages ID queries from exiting in earlier layers.

(2) *Information Bottleneck (IB) Regularization*: Earlier layers may lack compressive feature information, rendering those features not suitable for predicting the class label. To tackle this challenge, we introduce the following IB learning objective for ID training data. Let $I(Z_i; Y_{id})$ represent the mutual information between the two random variables, the latent features $Z_i$ and data labels $Y_{id}$. Similarly, $I(X_{id}; Z_i)$ denotes the mutual information between the two random variables, input ID data $X_{id}$ and latent features $Z_i$. In Eq. (3), $I(X_{id}; Z_i) - I(Z_i; Y_{id})$ is the IB loss. We maximize the mutual information between $Z_i$ and $Y_{id}$ to enhance the correlation between them to improve generalization. We minimize the mutual information between $X_{id}$ and $Z_i$ to reduce the effect of nuisance information of input $X_{id}$ on the features $Z_i$.

**OOD Data Learning Objective** Conversely, we put forth the following objectives for OOD queries:

$$\mathcal{L}_{ood} = \sum_{i=1}^{i=N} [I(Z_i; Y_{ood}) - I(X_{ood}; Z_i) + \alpha_i J(i) H(V_i^*(\boldsymbol{x}_{ood}))] \tag{4}$$

In Eq. (4), there are two distinct loss terms incorporated: entropy regularization and Information Bottleneck (IB) regularization. It is important to note that the learning objective for OOD data is opposite to the case in ID data. Subsequently, we elaborate on each of these loss terms.

(1) *Entropy Regularization*: $H(V_i^*(\boldsymbol{x}_{ood}))$ is the entropy regularization defined in Eq. (2), $\alpha_i$ is the hyperparameter for the $i^{th}$ exit, and the piecewise function $J(i)$ assigns values based on the comparison of the integer $i$ with the constant $N$. $J(i)$ returns a value of $-1$ when $i$ is equal to the $N$. If $i$ is less than the $N$, $J(i)$ instead returns a value of $1$. The aim is to maximize entropy at later layer, promoting a flat and uniform probability distribution to prevent OOD data from exiting at later layer. At the same time, we minimize entropy at the other layers, facilitating the output exit at the earlier layers, as lower entropy indicates larger probabilities concentrated on one class.

(2) *Information Bottleneck (IB) Regularization*: Certain earlier layers may retain some informative features crucial for predicting the class probabilities of OOD data. To reduce the correlations between OOD data features and labels and to encourage more random exit of OOD data at earlier layers, we optimize the *opposite* IB learning objective for OOD data as follows: Here, $I(Z_i; Y_{ood})$ represents the mutual information between latent features $Z_i$ and data labels $Y_{ood}$; $I(X_{ood}; Z_i)$ denotes the mutual information between OOD data $X_{ood}$ and latent features $Z_i$; In Eq. (4), $I(Z_i; Y_{ood}) - I(X_{ood}; Z_i)$ is the negative IB loss. This optimization is contrary to the IB optimization applied to the ID data. We minimize the mutual information between $Z_i$ and $Y_{ood}$ to diminish the correlation between features and predictions, thereby promoting a more random behavior in exit classifiers. Simultaneously, we maximize the mutual information between $X_{ood}$ and $Z_i$ to amplify the influence of the nuisance information from input $X_{ood}$ on the features $Z_i$, fostering a stronger defense mechanism.

## 4.2 Training Algorithm of DNF

While we formulate our learning objective in Section 4.1, calculating the mutual information is intractable in general since the mutual information involves calculating an expectation, which often requires integrating over the joint distribution of the variables involved. In many cases, this integration is computationally or analytically intractable. Variational inference provides a framework for approximating these intractable integrals. We derive the variational lower bound as the following:

$$I(Z_i; Y) - I(Z_i; X) \geq \int d\boldsymbol{x} d\boldsymbol{y} d\boldsymbol{z}_i P(\boldsymbol{x}) P(\boldsymbol{y}|\boldsymbol{x}) P(\boldsymbol{z}_i|\boldsymbol{x}) \log P(\boldsymbol{y}|\boldsymbol{z}_i) \tag{5}$$

$$- \int d\boldsymbol{x} d\boldsymbol{z}_i P(\boldsymbol{x}) P(\boldsymbol{z}_i|\boldsymbol{x}) \log \frac{P(\boldsymbol{z}_i|\boldsymbol{x})}{r(\boldsymbol{z}_i)}$$

where $r(\boldsymbol{z}_i)$ is a variational marginal approximation of the intractable marginal $P(\boldsymbol{z}_i)$ and is set to be $r(\boldsymbol{z}_i) = \mathcal{N}(0, I)$. We put the derivation steps in Appendix B.

To efficiently calculate Eq. (5), the $P(\boldsymbol{x}, \boldsymbol{y}) = P(\boldsymbol{x})P(\boldsymbol{y}|\boldsymbol{x})$ is approximated by empirical data distribution $P(\boldsymbol{x}, \boldsymbol{y}) = \frac{1}{N} \sum_{n=1}^{N} \delta_{\boldsymbol{x}_n}(\boldsymbol{x}) \delta_{\boldsymbol{y}_n}(\boldsymbol{y})$, $\delta_{\boldsymbol{x}_n}$ is the Dirac delta function on $\boldsymbol{x}_n$ and $\delta_{\boldsymbol{y}_n}$ is the Dirac delta function on $\boldsymbol{y}_n$. Assuming the EE network feature extractor has the form $P(\boldsymbol{z}_i|\boldsymbol{x}) = \mathcal{N}(\boldsymbol{z}_i | f_c^{\mu}(\boldsymbol{x}), f_c^{\Sigma}(\boldsymbol{x}))$. An MLP $f_c$ outputs the $K$-dimensional mean $\mu$ of $\boldsymbol{z}_i$ and the $K \times K$ covariance matrix $\Sigma$. Then we can use the reparameterization trick (Kingma & Welling, 2013; Alemi et al., 2017) to rewrite the $P(\boldsymbol{z}|\boldsymbol{x})d\boldsymbol{z} = P(\boldsymbol{\epsilon})d\boldsymbol{\epsilon}$, where $\boldsymbol{z} = f(\boldsymbol{x}, \boldsymbol{\epsilon})$ represents a deterministic function of $\boldsymbol{x}$ and a Gaussian random variable $\boldsymbol{\epsilon}$. This enables the distribution $P(\boldsymbol{z}|\boldsymbol{x})$ to be reparameterized as a function of $\boldsymbol{\epsilon}$. We can calculate the KL divergence between the $P(\boldsymbol{z}|\boldsymbol{x})$ and $r(\boldsymbol{z})$, and combine all together to minimize the empirical IB loss function as the following Eq. (6):

$$\mathcal{L}_{id} = \frac{1}{N} \sum_{i=1}^{i=N} \mathbb{E}_{\boldsymbol{\epsilon}}[-\log q(y_{id}|V_i^*(\boldsymbol{x}_{id}, \boldsymbol{\epsilon}))] + \mathbb{KL}(P(Z|\boldsymbol{x}_{id})|\beta(Z)) + \alpha_i K(i) H(V_i^*(\boldsymbol{x}_{id}, \boldsymbol{\epsilon})) \quad (6)$$

In Eq. (6), $\boldsymbol{\epsilon} \sim \mathcal{N}(0, I)$, the first term is to maximize the log-likelihood on ID data and the second term minimizes the KL divergence w.r.t to $\beta(Z)$ on ID data to make the latent features more compressive and extract better representations.

Conversely, the empirical IB loss function for OOD data can be summarized as the following:

$$\mathcal{L}_{ood} = \frac{1}{N} \sum_{i=1}^{i=N} \mathbb{E}_{\boldsymbol{\epsilon}}[\log q(y_{ood}|V_i^*(\boldsymbol{x}_{ood}, \boldsymbol{\epsilon}))] - \mathbb{KL}(P(Z|\boldsymbol{x}_{ood})|\beta(Z)) + \alpha_i J(i) H(V_i^*(\boldsymbol{x}_{ood}, \boldsymbol{\epsilon})) \quad (7)$$

where in Eq. (7), $\boldsymbol{\epsilon} \sim \mathcal{N}(0, I)$, the first term is to minimize the log likelihood on OOD data and the second term is to maximize the KL divergence w.r.t $\beta(Z)$ on OOD data to make the latent features more diverse so that the their exit points are more diverse, achieving effective defense and protecting the architecture from extracting. The overall learning objective is to minimize the following loss:

$$\mathcal{L}_d = \mathcal{L}_{id} + \mathcal{L}_{ood} \quad (8)$$

We summarize the joint training details in Algorithm 1 in Appendix. In line 3-4, we randomly sample ID and simulated OOD dataset. In line 5-8, we calculate the base DNF training loss by Eq. (6, 7, 8), respectively. Then, we update the Early-Exit neural network $V^*$ via SGD optimizer with respect to the exit classifiers parameters $\boldsymbol{\delta}_{V^*}$.

**Deployment of DNF** During testing, given the input query $\boldsymbol{x}$ and the array of confidence thresholds $\boldsymbol{r} = \{r_0, r_1, .., r_N\}$, where $r_0$ represents the first confidence threshold for the first intermediate exit classifier, and so forth. Starting from the input $\boldsymbol{x}$ and progressing to the position of the first intermediate exit classifier in the model $V^*$, the first intermediate exit classifier calculates the softmax probabilities $V_0^*(\boldsymbol{x})$ across all prediction classes. We denote the probability of the $j^{th}$ class in the prediction at the first intermediate exit classifier as $V_{0,j}^*(\boldsymbol{x})$. The largest probability value in $V_0^*(\boldsymbol{x})$ is denoted as $V_{0,max}^*(\boldsymbol{x}) = \max_j V_{0,j}^*(\boldsymbol{x})$. If $V_{0,max}^* \geq r_0$, it signifies that the model $V^*$ is confident in the current result, enabling the early termination of subsequent calculations. If $V_{0,max}^* < r_0$, the inference process continues. We summarize the DNF testing algorithm in Algorithm 2 in Appendix.

## 5 EXPERIMENTS

### 5.1 EXPERIMENTAL SETUP

**Datasets** (1) We evaluate the effectiveness of our method against DFME attack by using MNIST (10 classes) (Deng, 2012), CIFAR-10 (10 classes), CIFAR-100 datasets (100 classes) (Krizhevsky, 2009), and ImageNet-100 (Vinyals et al., 2016) (100 classes), as these datasets are commonly used in existing DFME research. (2) For evaluating the effectiveness of our method against DBME, following Mazeika et al. (2022), we use Caltech256 (Griffin et al., 2007) as the query dataset for both ImageNet200 (200 classes) and CUB200 (Wah et al., 2011) datasets trained victim models.

**Baselines** We evaluate the superiority of our method against five different strong defense baselines: (1) *Undefended*: without any defense. (2) Random Perturb (*RandP*) (Orekondy et al., 2020). (3) P-poison (Orekondy et al., 2020). (4) GRAD (Mazeika et al., 2022) (5) MeCo (Wang et al., 2023). We put a detailed description of these baselines in Appendix C.

**Implementation Details** In our experiments, following (Wang et al., 2023), in DFME attack, the $l_1$ perturbation budget is set to 1.0, for the defense baselines in order to mount a strong defense against

model extraction. This means that the $l_1$ norm of the difference between $y$ and $\hat{y}$, where $y$ represents the original output probabilities and $\hat{y}$ represents the modified output, does not exceed 1.0. In DFME attack setting, following (Truong et al., 2021), the query budget for different datasets we set is as follows, $2M$ for MNIST, $20M$ for CIFAR10, $200M$ for CIFAR100, $200M$ for ImageNet-100. In DBME attack, the query budget is set to be $10K$ for ImageNet200, $23K$ for CUB200. We report the results with a mean and standard deviation with five runs. All experiments are run on a single NVIDIA RTX A6000 GPU. Due to space limitations, we put the details of training and hyperparameter in Appendix D, exit classifier architecture in Appendix E, and OOD generator in F, respectively. *Exit Threshold Selection Guideline*: We propose a sample-efficient Bayesian optimization framework for automatically selecting the exit threshold. The detailed selection process is outlined in Appendix H.

**Early-Exit Victim Model** For MNIST dataset, LeNet-5 is the backbone model. For CIFAR10&100 and ImageNet-100 datasets, ResNet34-8x (He et al., 2016) is the backbone model. For CUB200 dataset, ResNet50 is the backbone model. For ImageNet200 dataset, Swin ViT (Liu et al., 2021) is the backbone model. During the DNF training, we freeze backbone parameters and only train exit classifiers. We put the victim model training details in Appendix G.

## 5.2 Defense against Data-free Model Extraction

**DFME attack baselines** We adopt the following DFME attack baselines: the soft-label attack method DFME (Truong et al., 2021), and the hard-label attack method DFMS-HL (Sanyal et al., 2022a).

**Results on CIFAR10 and CIFAR100** The results of defense against soft-label and hard-label DFME attack on CIFAR10 and CIFAR100 are shown in Table 1. Our EENN's backbone is based on ResNet34. We use three distinct model architectures as the clone model architectures, which include ResNet-18 (He et al., 2016), MobileNetV2 (Sandler et al., 2018) and DenseNet121 (Huang et al., 2017). Compared to the undefended method, under soft-label attack setting, DNF can significantly reduce clone model accuracy by $27\%$ to $34\%$ on the CIFAR10 dataset and by $17\%$ to $41\%$ on the CIFAR100 dataset. Under hard-label attack setting, DNF can significantly reduce clone model accuracy by $4\%$ to $8\%$ on CIFAR10 and by $14\%$ to $19\%$ on CIFAR100. The clone model accuracy under different defense methods with varying query budgets is shown in Figure 2 in Appendix I.

Table 1: Clone model accuracy on **CIFAR-10** and **CIFAR-100** with *ResNet34* as the victim model.

| Attack | Defense | **CIFAR-10** Clone Model Architecture | | | **CIFAR-100** Clone Model Architecture | | |
|---|---|---|---|---|---|---|---|
| | | ResNet18-8X | MobileNetV2 | DenseNet121 | ResNet18-8X | MobileNetV2 | DenseNet121 |
| DFME (Soft-label) | Undefended ↓ | $87.36 \pm 0.78\%$ | $75.23 \pm 1.53\%$ | $73.89 \pm 1.29\%$ | $58.72 \pm 2.82\%$ | $28.36 \pm 1.97\%$ | $27.28 \pm 2.08\%$ |
| | RandP ↓ | $84.28 \pm 1.37\%$ | $70.56 \pm 2.23\%$ | $70.03 \pm 2.38\%$ | $41.69 \pm 2.91\%$ | $22.75 \pm 2.19\%$ | $23.61 \pm 2.70\%$ |
| | P-poison ↓ | $78.06 \pm 1.73\%$ | $66.32 \pm 1.36\%$ | $68.75 \pm 1.40\%$ | $38.72 \pm 3.06\%$ | $20.87 \pm 2.61\%$ | $21.89 \pm 2.93\%$ |
| | GRAD ↓ | $79.33 \pm 1.68\%$ | $65.82 \pm 1.67\%$ | $69.06 \pm 1.57\%$ | $39.07 \pm 2.72\%$ | $20.71 \pm 2.80\%$ | $22.08 \pm 2.78\%$ |
| | MeCo ↓ | $\mathbf{51.68 \pm 1.96\%}$ | $46.53 \pm 2.09\%$ | $61.38 \pm 2.41\%$ | $29.57 \pm 1.97\%$ | $12.18 \pm 1.05\%$ | $10.79 \pm 1.36\%$ |
| | DNF ↓ | $53.91 \pm 2.30\%$ | $\mathbf{46.32 \pm 1.45\%}$ | $\mathbf{47.21 \pm 2.15\%}$ | $\mathbf{18.03 \pm 3.03\%}$ | $\mathbf{10.82 \pm 1.34\%}$ | $\mathbf{6.75 \pm 1.23\%}$ |
| DFMS-HL (Hard-label) | Undefended ↓ | $84.67 \pm 1.90\%$ | $79.28 \pm 1.87\%$ | $68.87 \pm 2.08\%$ | $72.57 \pm 1.28\%$ | $62.71 \pm 1.68\%$ | $63.58 \pm 1.79\%$ |
| | RandP ↓ | $84.02 \pm 2.31\%$ | $78.71 \pm 1.93\%$ | $68.16 \pm 2.23\%$ | $72.43 \pm 1.43\%$ | $62.06 \pm 1.82\%$ | $63.16 \pm 1.73\%$ |
| | P-poison ↓ | $84.06 \pm 1.87\%$ | $79.02 \pm 1.96\%$ | $68.05 \pm 2.17\%$ | $71.83 \pm 1.32\%$ | $61.83 \pm 1.79\%$ | $62.73 \pm 1.91\%$ |
| | GRAD ↓ | $84.28 \pm 1.95\%$ | $78.83 \pm 1.91\%$ | $68.11 \pm 1.93\%$ | $71.89 \pm 1.37\%$ | $62.60 \pm 1.71\%$ | $62.57 \pm 1.80\%$ |
| | MeCo ↓ | $76.86 \pm 2.09\%$ | $\mathbf{71.22 \pm 1.87\%}$ | $62.33 \pm 2.01\%$ | $59.30 \pm 1.70\%$ | $55.32 \pm 1.65\%$ | $56.80 \pm 1.86\%$ |
| | DNF ↓ | $\mathbf{76.51 \pm 2.12\%}$ | $75.01 \pm 1.25\%$ | $\mathbf{61.02 \pm 1.21\%}$ | $\mathbf{52.98 \pm 2.24\%}$ | $\mathbf{48.41 \pm 1.78\%}$ | $\mathbf{49.72 \pm 1.24\%}$ |

For **MNIST** and **ImageNet-100** dataset, we put detailed results in Appendix J.1.

**Model Utility Evaluation.** We evaluate the victim model utility with various defense strategies in Table 4. We can see the DNF maintains high test accuracy compared with other defense strategies. This proves that our DNF method not only improves the defense capability and computational efficiency, but also improves the victim model utility for legitimate users. We evaluate the accuracy of ID data at each exit on CIFAR10 dataset. We use 10,000 test samples and force predictions to occur at a single exit, as shown in Figure 3 in the Appendix, the model achieves progressively higher accuracy at later exit points, with corresponding error bars indicating the standard deviation.

## 5.3 Defense against Data-based Model Extraction

To evaluate the effectiveness of DNF under DBME attack, we follow the random strategy in Knockoff Nets (Orekondy et al., 2019). The attacker randomly samples images from a different distribution than that of ID data to extract the victim model. For uncertainty-based sampling as proposed in ActiveThief (Pal et al., 2020), we provide the results in Table 19 in the Appendix.

**Results on CUB200 and ImageNet200** To verify the effectiveness of DNF on CUB200 and ImageNet200 datasets, our EENN's backbone is ResNet50 and Swin Transformer, respectively. We

choose ResNet50, ResNet34, and VGG19 as the clone model architectures for CUB200 dataset and Vision Transformer (ViT) (Dosovitskiy et al., 2020) (Large), CaiT (Touvron et al., 2021b), DeiT (Touvron et al., 2021a) and Swin (Liu et al., 2021) (Large) for ImageNet200 dataset. The results are shown in Table 2. Compared to the undefended method, under soft-label attack setting, DNF can significantly reduce clone model accuracy by 31% to 40% on the CUB200 dataset and by 11% to 19% on the ImageNet200 dataset. Under hard-label attack setting, DNF can significantly reduce clone model accuracy by 10% to 15% on the CUB200 dataset and by 12% to 16% on ImageNet200.

Table 2: Clone model accuracy on **CUB200** and **ImageNet200** with *ResNet50* as the victim model.

| Attack | Defense | **CUB200** Clone Model Architecture | | | **ImageNet200** Clone Model Architecture | | | |
|---|---|---|---|---|---|---|---|---|
| | | ResNet50 | ResNet34 | VGG19 | ViT-Large | CaiT | DeiT | Swin-Large |
| Soft-label | Undefended ↓ | $55.49 \pm 1.50\%$ | $36.71 \pm 0.79\%$ | $39.85 \pm 1.71\%$ | $73.35 \pm 0.89\%$ | $62.26 \pm 1.24\%$ | $56.59 \pm 1.54\%$ | $60.67 \pm 1.43\%$ |
| | RandP | $39.77 \pm 0.35\%$ | $20.59 \pm 1.03\%$ | $24.02 \pm 0.35\%$ | $69.69 \pm 1.08\%$ | $59.58 \pm 1.78\%$ | $53.97 \pm 1.89\%$ | $56.83 \pm 0.98\%$ |
| | P-poison | $24.94 \pm 1.25\%$ | $15.31 \pm 1.17\%$ | $20.81 \pm 1.33\%$ | $65.53 \pm 1.16\%$ | $58.91 \pm 1.74\%$ | $52.08 \pm 0.53\%$ | $55.67 \pm 1.58\%$ |
| | GRAD | $24.32 \pm 0.55\%$ | $15.06 \pm 1.11\%$ | $20.65 \pm 0.24\%$ | $65.32 \pm 0.68\%$ | $59.24 \pm 0.75\%$ | $51.97 \pm 1.82\%$ | $55.79 \pm 0.57\%$ |
| | MeCo | $51.32 \pm 0.54\%$ | $31.98 \pm 0.73\%$ | $34.36 \pm 0.25\%$ | $69.93 \pm 0.32\%$ | $60.17 \pm 0.20\%$ | $53.78 \pm 0.79\%$ | $58.85 \pm 0.73\%$ |
| | DNF ↓ | $\mathbf{15.32 \pm 1.21\%}$ | $\mathbf{5.27 \pm 1.72\%}$ | $\mathbf{8.21 \pm 2.23\%}$ | $\mathbf{56.03 \pm 1.12\%}$ | $\mathbf{46.21 \pm 0.91\%}$ | $\mathbf{45.31 \pm 1.09\%}$ | $\mathbf{42.10 \pm 1.26\%}$ |
| Hard-label | Undefended ↓ | $31.29 \pm 1.58\%$ | $21.57 \pm 0.62\%$ | $23.27 \pm 0.80\%$ | $63.57 \pm 0.69\%$ | $57.73 \pm 0.80\%$ | $53.16 \pm 1.52\%$ | $60.26 \pm 1.23\%$ |
| | RandP | $30.89 \pm 0.61\%$ | $21.68 \pm 0.91\%$ | $23.44 \pm 1.34\%$ | $63.18 \pm 1.14\%$ | $57.31 \pm 0.80\%$ | $52.86 \pm 0.45\%$ | $59.58 \pm 1.01\%$ |
| | P-poison | $30.69 \pm 0.91\%$ | $21.54 \pm 1.98\%$ | $22.06 \pm 1.01\%$ | $63.09 \pm 0.65\%$ | $57.12 \pm 0.51\%$ | $52.57 \pm 1.12\%$ | $59.23 \pm 1.87\%$ |
| | GRAD | $31.23 \pm 0.61\%$ | $22.38 \pm 1.52\%$ | $22.37 \pm 0.44\%$ | $63.21 \pm 1.24\%$ | $57.23 \pm 1.90\%$ | $52.34 \pm 1.54\%$ | $59.30 \pm 1.23\%$ |
| | MeCo | $29.42 \pm 0.46\%$ | $19.90 \pm 0.42\%$ | $21.08 \pm 0.34\%$ | $63.31 \pm 0.48\%$ | $57.25 \pm 0.24\%$ | $52.69 \pm 0.45\%$ | $59.61 \pm 0.50\%$ |
| | DNF ↓ | $\mathbf{16.01 \pm 1.24\%}$ | $\mathbf{10.91 \pm 1.02\%}$ | $\mathbf{12.98 \pm 1.25\%}$ | $\mathbf{48.95 \pm 2.11\%}$ | $\mathbf{43.65 \pm 0.91\%}$ | $\mathbf{40.98 \pm 1.69\%}$ | $\mathbf{44.01 \pm 0.82\%}$ |

Table 3: Effect of different OOD datasets on defense performance during DNF training on **ImageNet200** with *Swin Transformer* as the victim model.

| Setting | Undefended ↓ | DNF (Indoor67) ↓ | DNF (SVHN) ↓ | DNF (GAN) ↓ |
|---|---|---|---|---|
| Soft-label | $73.35 \pm 0.89\%$ | $56.03 \pm 1.12\%$ | $\mathbf{55.51 \pm 1.81\%}$ | $59.71 \pm 1.71\%$ |
| Hard-label | $63.57 \pm 0.69\%$ | $48.95 \pm 2.11\%$ | $\mathbf{48.73 \pm 1.72\%}$ | $49.98 \pm 1.91\%$ |

Table 4: **Victim model utility** measured by the test accuracy

| Method | CIFAR10 | CIFAR100 | CUB200 | ImageNet200 |
|---|---|---|---|---|
| Undefended ↑ | $94.91 \pm 0.37\%$ | $76.71 \pm 1.25\%$ | $81.18 \pm 1.21\%$ | $91.55 \pm 0.32\%$ |
| RandP ↑ | $93.98 \pm 0.28\%$ | $75.23 \pm 1.39\%$ | $80.58 \pm 0.12\%$ | $91.37 \pm 0.45\%$ |
| P-poison ↑ | $94.58 \pm 0.61\%$ | $75.42 \pm 1.21\%$ | $80.20 \pm 0.34\%$ | $91.24 \pm 0.17\%$ |
| GRAD ↑ | $94.65 \pm 0.67\%$ | $75.60 \pm 1.45\%$ | $80.23 \pm 0.15\%$ | $91.28 \pm 0.32\%$ |
| MeCo ↑ | $94.17 \pm 0.56\%$ | $75.36 \pm 0.68\%$ | $79.42 \pm 0.53\%$ | $90.69 \pm 0.58\%$ |
| DNF ↑ | $94.34 \pm 0.07\%$ | $78.73 \pm 0.30\%$ | $80.13 \pm 0.86\%$ | $91.50 \pm 0.24\%$ |

## 5.4 IN-DISTRIBUTION JBDA DEFENSE AND ARCHITECTURE STEALING DEFENSE

**Defense Against JBDA with In-Distribution Data Attack:** To assess the effectiveness of DNF when the attacker has access to in-distribution data for queries, we tested its performance under the JBDA (Papernot et al., 2017; Juuti et al., 2019) attack. As shown in Table 17, DNF achieves the lowest clone model accuracy of 40.60% in the soft-label attack. In the hard-label attack, it reduces the clone accuracy to 44.20%, outperforming other defense methods. Overall, DNF consistently demonstrates robust performance in lowering clone model accuracy across both attack scenarios.

**Defense against Model Architecture Stealing:** To assess the effectiveness of our defense against model architecture stealing, as proposed in (Carlini et al., 2024), we use a pre-trained ViT-Large model with 200 output classes on the ImageNet200 dataset as the victim model. The objective is to steal the hidden dimension prior to the class logits output. We set the hidden dimensions for different exit classifiers to be [80, 100, 120, 140, 160]. Using 300 images from the ImageNet200 dataset, we query the victim model and obtain an output matrix $\mathcal{O}^{300 \times 200}$. Following the method described in (Carlini et al., 2024), we calculate and sort the singular values of $\mathcal{O}$ as $\lambda_1 \geq \lambda_2 \geq \cdots \geq \lambda_n$. The index that produces the largest difference between consecutive singular values is selected as the hidden dimension. The output of the architecture stealing algorithm in (Carlini et al., 2024) identified 137 as the hidden dimension, which is entirely different from the hidden dimensions set in our exit classifiers. This discrepancy demonstrates the effectiveness of our defense in protecting the model architecture from being stolen.

## 5.5 ADAPTIVE ATTACKS

We further evaluate the robustness of DNF against potential countermeasures by attackers. We proposed two types of adaptive attacks. Specifically, **(1)** we consider the DFME soft-label attack, where attackers are aware of our defense strategy, including the early-exit network architecture and loss function, referred to as the Adaptive Entropy Attack. In this scenario, attackers employ adaptive attacks by incorporating the same designed entropy loss as our DNF into the clone model's loss function, updating the clone model together to attempt bypassing our defense. **(2)** We also evaluate an EE Adaptive Entropy Attack, where attackers have even more information, using the same early-exit architecture as the victim model. As shown in Appendix Tables 9, DNF is still effective against these two adaptive attacks.

## 5.6 ABLATION STUDY

**Exit Point Threshold Evaluation** We evaluate the influence of the threshold choices for each exit point. As shown in Appendix Table 10, the threshold selection does not affect the defense ability of DNF too much. We also propose a sample-efficient Bayesian optimization framework to automatically select the optimal exit threshold in Appendix H.

**Different OOD Dataset Evaluation** We assess the impact of using different OOD datasets for DNF training. For instance, as demonstrated in Table 3, DNF consistently reduces clone model accuracy across different OOD datasets, including Indoor67, SVHN and GAN-generated, with small variation in clone model accuracy. This highlights the stability of DNF regardless of the OOD dataset used.

**Inference Efficiency Evaluation Compared with Fixed Architecture** To assess the inference efficiency of dynamic network architecture as opposed to fixed architecture, we conduct a comparison of their respective efficiency in Appendix Table 11, which shows that DNF achieves a notable $2\times$ improvement in inference efficiency than the SOTA defense method and undefended model.

**Fine-tuning Efficiency** We measure the training time of DNF. As shown in Appendix Table 13 & 14, the wall-clock time of training is about 20 seconds for the MNIST dataset, the wall-clock time of training is about 7 hours for Swin Transformer, the fine-tuning is highly efficient.

**Exit Point Evaluation** To assess the exit point percentage of input query data on attack query and benign query data, we analyze the exit point percentages for attack and benign query data in Appendix Table 12, which reveals that $54.93\%$ of OOD data generated by the attacker is exiting from the initial exit point. Because the initial exit point typically doesn't generate a lot of meaningful information, which results in these OOD queries disrupting the training process of the attacker's model. Some OOD data exiting from later layers could be attributed to shifts in data distribution. Conversely, when processing the ID data, it is noteworthy that over half of the data exited through the third exit. This significantly reduces the time consumption for model inference. This can be attributed to the advantages of early-exit networks in avoiding overthinking, as highlighted by (Kaya et al., 2019), which also enhance inference efficiency for ID data. Overall, ID data tends to exit at relatively deeper layers compared to OOD data. A more detailed explanation of the ID and OOD exit percentages across different exit classifiers is provided in the Appendix above Table 12.

**The impact of different loss weights on the results** We measure if the different weights $\tau$ of ID and OOD losses $\mathcal{L}_{id} + \tau\mathcal{L}_{ood}$ in Eq. (8) will influence the final results. As shown in Appendix Table 15, the change in clone model accuracy is not large, which illustrates DNF's stability.

**The impact of different entropy weights $\alpha$ on the clone model accuracy** We measure the impact of different entropy weights $\alpha$ in Eq. (6) and (7) on the defense performance. As shown in Appendix Table 18, the OOD data loss term contribute significantly to the effectiveness of our approach. Despite variations in the entropy weight, the changes in defense performance is not large, demonstrating the stability of DNF across different $\alpha$.

**Effectiveness of DNF on protecting the architecture searched by neural architecture search (NAS)** In addition to the traditional network architectures, we also verified whether our method can be used on more general architectures searched by NAS (Lu et al., 2020). Our method is still effective as shown in Appendix Table 16.

## 6 CONCLUSION

We present DNF, a novel defense strategy aimed at efficiently countering model extraction attacks by achieving three key objectives: (1) safeguarding model functionality, (2) protecting the network architecture, and (3) improving defense efficiency. Our method steers attack queries to exit at earlier network layers for effective protection, while allowing benign queries to exit at later layers, thereby preserving model utility. Comprehensive experiments across various model extraction scenarios and datasets confirm the effectiveness and efficiency of our approach.

**Limitations** Our approach does not protect the entire network architecture from being stolen. Even though certain parts of the network are executed, the attacker cannot determine which inputs correspond to which specific exit classifiers. This uncertainty helps preserve the protection of the network architecture.

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

In this Appendix, we provide more details of Model Extraction Details in A, Derivations of the Information Bottleneck Lower Bound in B, the details of our experiment bases in C, the training and hyperparameter selection in Section D, exit classifier architecture in Section E, OOD Generator in Section F, the Victim Model Backbone Training in G, and the more the Performance evaluation of defense methods as budget increases in Section I, and the more results of experiments in Section J.

# A  MODEL EXTRACTION DETAILS

**Soft-label Attack** In a soft-label DFME attack, a generative model $G$ is employed to generate input data. Initially, attacker samples a random noise vector $\epsilon \sim \mathcal{N}(0, I)$, which is then input into the generative model $G$ to produce an image $x$, represented as $x = G(\epsilon, \delta_G)$. Subsequently, a clone model $C$ is trained to align its predictions with those of the victim model $V$ on the input data $x$, expressed as the following:

$$\min_{\delta_C} \mathbb{KL}(C(x, \delta_C) || V(x, \delta_V)) \tag{9}$$

**Hard-label Attack** In a hard-label DFME attack Sanyal et al. (2022b), a generative adversarial network (GAN) Goodfellow et al. (2014)-based architecture is trained to generate query data for querying the victim model and training the clone model $C$. Initially, a DCGAN Radford et al. (2016) is trained by imposing an image prior using synthetic or unrelated proxy data, serving as the initial training phase for the generator $G$. Subsequently, the clone model $C$ and the generator $G$ undergo alternating training. The generator $G$ synthesizes data $x$ through $x = G(\epsilon, \delta_G)$, where $\epsilon \sim \mathcal{N}(0, I)$. The victim model $V$ processes $x$ and delivers the hard-label $\hat{y}(x)$ to the attacker. The attacker constructs input-label pairs $(x, \hat{y}(x))$ using this information, which are then utilized to train the clone model $C$. Additionally, the generated data $x$ is used to train the generator $G$ through adversarial and diversity losses.

# B  DERIVATIONS OF THE INFORMATION BOTTLENECK LOWER BOUND

Given $I(Z_i; Y)$ and $I(Z_i; X)$, we have the following derivation.

$$I(Z_i, Y) \geq \int dx \, dy \, dz_i \, P(x)P(y|x)P(z_i|x) \log q(y|z_i). \tag{10}$$

$$I(Z_i, X) = \int dz_i \, dx \, P(x, z_i) \log \frac{P(z_i|x)}{P(z_i)} = \int dz_i \, dx \, P(x, z_i) \log P(z_i|x) - \int dz_i \, P(z_i) \log P(z_i) \tag{11}$$

We can use $\beta(Z)$ be a variational approximation to

$$P(z_i) = \int dx \, P(z_i|x) \, P(x) \tag{12}$$

Based on Eq. (12), Eq. (11) can be rewritten as (13)

$$I(Z_i, X) \leq \int dx \, dz_i \, P(x)P(z_i|x) \log \frac{P(z_i|x)}{r(z_i)}. \tag{13}$$

Based on (10) and (13), we have that:

$$I(Z_i; Y) - I(Z_i; X) \geq \int dxdydz_i \, P(x)P(y|x)P(z_i|x) \log q(y|z_i) \tag{14}$$

$$- \int dxdz_i \, P(x)P(z_i|x) \log \frac{P(z_i|x)}{r(z_i)} = L. \tag{15}$$

## C    BASELINE

- **RandP** Orekondy et al. (2020); Rigaki & Garcia (2023): the paper proposes to perturb the prediction vectors of the victim model, the goal is poisoning the training objective of the attacker.
- **Prediction Poisoning** Orekondy et al. (2020): the paper introduces a perturbation objective termed 'Maximizing Angular Deviation' (MAD). The core objective of MAD is to alter the prediction probabilities of the victim model. This alteration is strategically designed to produce an adversarially perturbed gradient, which diverges maximally from the original gradient of the victim model.
- **GRAD** Mazeika et al. (2022): the paper represents a gradient redirection method, designed to enable the adversarial gradient update in any arbitrary direction. The purpose of this method is to control the direction of the gradient update.
- **MeCo** Wang et al. (2023): the paper represents both memory and computation efficient defense method through distributionally robust defensive training by adding a data-dependent random perturbation generator to perturb the input data. The attacker cannot steal useful information from the black-box model, at the same time, MeCo can keep the target model utility.

## D    THE TRAINING AND HYPERPARAMETER DETAILS

When training the ResNet34-8x backbone for CIFAR10/100 datasets, we set 200 epochs in total, the initial learning rate is 0.1, optimizer is SGD. The learning rate will be adjusted as epochs grow. If the number of epochs is smaller than 80, the learning rate is 0.1, if the number of epochs is between 80 with 120, the learning rate is 0.01. If the number of epochs is bigger than 120, the learning rate is 0.001.

When training the DNF, we set the $\alpha = 0.1$ by default, the number of epochs is 10, and the optimizer is SGD, the learning rate is 0.001. For MNIST dataset, we set confidence thresholds array as $[0.92, 0.94, 0.96]$ by default. For CIFAR10/100, TinyImageNet, CUB200 and ImageNet200 datasets, we set confidence thresholds array as $[0.90, 0.92, 0.94, 0.96]$ by default. In our design, the default confidence thresholds are progressively increasing. Due to our loss function, this ensures that decisions at earlier exits are made with relatively lower confidence, allowing OOD data to exit earlier, while ID data progresses to later exits where decisions are made with higher confidence. Additionally, we opt for different thresholds across different exit layers for two reasons. First, as this approach enhances defense against adaptive attacks. It makes it more challenging for attackers to accurately predict the correct threshold for all exit layers simultaneously. Second, using different threshold values for each exit increases the separation between the exit distributions of ID and OOD data, further improving defense performance.

## E    THE ARCHITECTURE DETAILS OF EXIT CLASSIFIER

If $V$ is ResNet34-8x which contains 20 million trainable parameters, based on its structure features, we add the exit classifier after each Residual Blocks of victim model $V$, in total, excluding the original $V$ exit, we add four more exits $\{V_i^*\}_{i=0}^{i=3}$. The architecture details for ResNet34-8x based exit classifier as shown in Table 5, and all exit classifiers $\{V_i^*\}_{i=0}^{i=3}$ contains 100440 trainable parameters, in total, we add 0.47% trainable parameters, which is acceptable.

Table 5: The architecture details of exit classifiers, $M$ is the numbher of classes.

| Exit Index | First layer | Second layer | Parameters |
|---|---|---|---|
| Exit1 | $Linear(64, 100)$ | $Linear(100, M)$ | 7510 |
| Exit2 | $Linear(128, 100)$ | $Linear(100, M)$ | 13910 |
| Exit3 | $Linear(256, 100)$ | $Linear(100, M)$ | 26710 |
| Exit4 | $Linear(512, 100)$ | $Linear(100, M)$ | 52310 |

If $V$ is LeNet-5 which contains $61K$ trainable parameters, based on its structure features, we add the exit classifier after each convolution layer of victim model $V$, in total, excluding the original $V$ exit,

we add three more exits $\{V_i^*\}_{i=0}^{i=2}$. The architecture details for LeNet-5 based exit classifier as shown in Table 6, and all exit classifiers $\{V_i^*\}_{i=0}^{i=2}$ contains 16990 trainable parameters, in total, we add 27% trainable parameters, which is acceptable.

Table 6: The architecture details of exit classifiers, $M$ is the numbher of classes.

| Exit Index | First layer | Parameters |
|---|---|---|
| Exit1 | $Linear(1176, M)$ | 11770 |
| Exit2 | $Linear(400, M)$ | 4010 |
| Exit3 | $Linear(120, M)$ | 1210 |

The architecture details of exit classifiers for other networks, such as transformer-series are similar to this, the difference is we will use LayerNorm.

## F  OOD GENERATOR

For generating out-of-distribution (OOD) data, we use a Deep Convolutional Generative Adversarial Network (DCGAN) Radford et al. (2016). The training dataset for this DCGAN is the TinyImageNet (if the TinyImageNet is the ID dataset, we choose other datasets as the training dataset, such as SVHN), a widely recognized subset in the field of computer vision. We set the learning rate (lr) to 0.0002, and 30 epochs in total. Furthermore, we choose the Adam optimizer model training.

## G  VICTIM MODEL BACKBONE TRAINING

For LeNet-5 as the victim model for MNIST datasets, we run 20 epochs in total and set the initial learning rate as 0.001, optimizer as Adam. For ResNet34-8x He et al. (2016) as the victim model for CIFAR10&100 and ImageNet-100 datasets, we run 200 epochs in total and set the initial learning rate as 0.1, optimizer as SGD. For Swin Transformer (Swin) Liu et al. (2021) as the victim model for ImageNet200 datasets, we run 30 epochs in total and set the initial learning rate as 0.01, optimizer as Adam with decoupled weight decay (AdamW) Loshchilov & Hutter (2017). Furthermore, we follow the procedure in Truong et al. (2021); Tang et al. (2024) for the clone model training.

## H  BAYESIAN HYPERPARAMETER OPTIMIZATION FOR EXIT THRESHOLDS

We provide the following step-by-step guide to Bayesian Hyperparameter Optimization for Exit Thresholds Selection

**Step 1**: Early exit networks have multiple exits where predictions can be made. Each exit has an associated threshold (confidence level) that determines whether to exit or proceed to the next stage. Let's denote these thresholds as $\boldsymbol{r} = \{r_0, r_1, .., r_N\}$ for an $N$-exit network. The goal is to determine the optimal set of thresholds $\boldsymbol{r} = \{r_0, r_1, .., r_N\}$ that minimize the clone model's accuracy on the validation set of the target dataset. To ensure generalization, we select the exit thresholds using a different query or attack dataset than the one employed by the attacker. For example, if the victim model is trained on CIFAR10, the attacker may use ImageNet100 for extraction. To select the exit thresholds, we query the victim model with a third dataset, such as CIFAR100, ensuring no prior knowledge of the attack query data distribution. We then assess the clone model's accuracy on CIFAR10's validation set to evaluate the performance of the chosen thresholds. We employ the Bayesian Optimization framework, specifically SMAC3 (Lindauer et al., 2022), which leverages a surrogate model to approximate the objective function and efficiently guide the search for optimal exit thresholds.

**Step 2**: Set the search space for each threshold $r_i$ to be within the range [0.85,1]. Start with an initial set of samples to fit the surrogate model. The optimizer of SMAC3 will then iteratively select the next set of thresholds to evaluate by balancing exploration (trying new values) and exploitation (refining around known good values). For each set of thresholds $\boldsymbol{r} = \{r_0, r_1, .., r_N\}$ proposed by the optimizer, train the early exit network and measure the clone model accuracy on the validation set of the target dataset. Return the result to the Bayesian optimizer of SMAC3 and update the surrogate model. Once the optimization process is complete, select the set of exit thresholds $\boldsymbol{r} = \{r_0, r_1, .., r_N\}$ that

yielded the best performance according to the objective function. Furthermore, we opt for different thresholds across different exit layers for two reasons. First, as this approach enhances defense against adaptive attacks. It makes it more challenging for attackers to accurately predict the correct threshold for all exit layers simultaneously. Second, using different threshold values for each exit increases the separation between the exit distributions of ID and OOD data, further improving defense performance.

## I  Performance evaluation of defense methods as budget increases

As shown in Figure 2, as budgets grow, the performance of clone models continues to improve, but DNF's defense performance is always better than other methods.

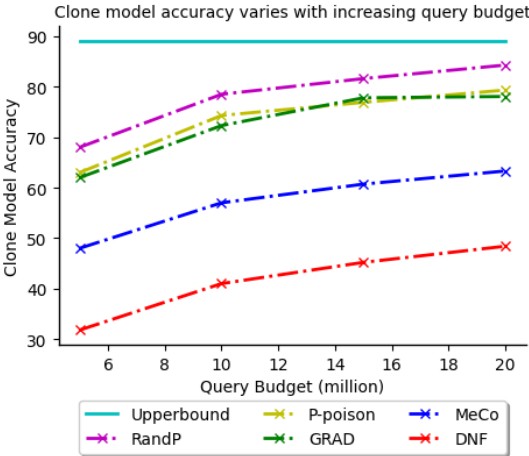

Figure 2: Performance of cloning models as budget increases for different defense methods. The plot shows the relationship between clone model accuracy and increasing query budget (in millions) under different defense methods. The x-axis represents the query budget, ranging from 6 to 20 million, while the y-axis shows the clone model accuracy, which increases as the query budget grows. The graph compares several defense strategies, including RandP, P-poison, GRAD, MeCo, and DNF. DNF consistently yields the lowest clone model accuracy across all query budgets.

## J  More Experiment Results

### J.1  Results on MNIST and ImageNet-100

**Results on MNIST** For MNIST dataset, under soft-label attack setting, compared to the undefended method, the results as shown in Table 7 that DNF can significantly reduce the effectiveness of existing soft-label DFME attack by up to $42\%$ when the clone model is LeNet5, $50\%$ when the clone model is LeNet-Half.

**Results on ImageNet-100** For ImageNet-100 dataset, under soft-label attack setting, compared to the undefended method, the results as shown in Table 8 that DNF can significantly reduce the effectiveness of existing soft-label DFME attack by up to $25\%$ when the clone model is ResNet-18, $22\%$ when the clone model is MobileNetV2, and $17\%$ when the clone model is DenseNet121.

Table 7: Clone model accuracy on **MNIST** with *LeNet5* as the victim model.

| Attack | Defense | Clone Model Architecture | |
|---|---|---|---|
| | | LeNet5 | LeNet5-Half |
| DFME (Soft-label) | Undefended ↓ | $98.76 \pm 0.27\%$ | $96.65 \pm 0.43\%$ |
| | RandP ↓ | $92.25 \pm 0.32\%$ | $91.86 \pm 0.49\%$ |
| | P-poison ↓ | $88.34 \pm 0.78\%$ | $86.09 \pm 0.96\%$ |
| | GRAD ↓ | $87.22 \pm 0.70\%$ | $85.38 \pm 0.91\%$ |
| | MeCo ↓ | $85.07 \pm 0.87\%$ | $82.93 \pm 1.27\%$ |
| | DNF ↓ | $\mathbf{55.97 \pm 3.19\%}$ | $\mathbf{46.26 \pm 5.78\%}$ |

Table 8: Clone model accuracy on **ImageNet-100** with *ResNet34-8x* as the victim model.

| Attack | Defense | Clone Model Architecture | | |
|---|---|---|---|---|
| | | ResNet18-8X | MobileNetV2 | DenseNet121 |
| DFME (Soft-label) | Undefended ↓ | $35.89 \pm 3.97\%$ | $28.71 \pm 3.25\%$ | $25.05 \pm 3.68\%$ |
| | RandP ↓ | $30.76 \pm 4.09\%$ | $22.06 \pm 3.83\%$ | $20.23 \pm 3.97\%$ |
| | P-poison ↓ | $29.36 \pm 4.23\%$ | $21.83 \pm 3.77\%$ | $20.01 \pm 3.89\%$ |
| | GRAD ↓ | $29.87 \pm 3.76\%$ | $21.65 \pm 3.75\%$ | $19.82 \pm 3.77\%$ |
| | MeCo ↓ | $23.29 \pm 3.83\%$ | $17.83 \pm 3.67\%$ | $16.73 \pm 3.88\%$ |
| | DNF (Ours) ↓ | $\mathbf{10.21 \pm 0.65\%}$ | $\mathbf{6.78 \pm 0.92\%}$ | $\mathbf{7.87 \pm 1.23\%}$ |

## J.2 ABLATION STUDY

Table 9: Clone model accuracy on **CIFAR-10** with *ResNet34-8x* as the victim model under **Adaptive Attack**.

| Attack | Defense | Accuracy |
|---|---|---|
| Original Soft-label Attack | Undefended ↓ | $87.36 \pm 0.78\%$ |
| Adaptive Entropy Attack | DNF ↑ | $\mathbf{57.62 \pm 2.24\%}$ |
| EE Adaptive Entropy Attack | DNF ↑ | $\mathbf{59.07 \pm 2.03\%}$ |

Table 10: **Exit Point Threshold Effect** on clone model accuracy on **CIFAR-10** with *ResNet34-8x* as the victim model under Original Soft-label Attack.

| Method | Clone model accuracy | Victim model accuracy |
|---|---|---|
| [0.90, 0.90, 0.90, 0.90] | $58.62 \pm 1.25\%$ | $94.50 \pm 0.19\%$ |
| [0.91, 0.91, 0.91, 0.91] | $57.31 \pm 2.12\%$ | $94.78 \pm 0.22\%$ |
| [0.90, 0.92, 0.94, 0.96] | $53.91 \pm 2.30\%$ | $94.34 \pm 0.07\%$ |
| [0.95, 0.95, 0.95, 0.95] | $56.53 \pm 2.19\%$ | $95.21 \pm 1.23\%$ |

Table 11: Running time (seconds) during deployment

| Algorithm | CIFAR10 | CIFAR100 |
|---|---|---|
| Undefended | $10.12 \pm 1.83$ | $10.96 \pm 2.09$ |
| P-poison | $382.49 \pm 2.78$ | $1926.88 \pm 8.71$ |
| GRAD | $183.47 \pm 1.66$ | $543.49 \pm 4.29$ |
| MeCo | $10.36 \pm 0.80$ | $11.17 \pm 0.82$ |
| DNF_ID | $\mathbf{6.42 \pm 2.54}$ | $\mathbf{6.89 \pm 2.51}$ |
| DNF_OOD | $\mathbf{5.41 \pm 2.21}$ | $\mathbf{5.93 \pm 2.32}$ |

In Table 12, we observe that the majority of ID data exits at the third exit, with some exiting at earlier points. Similarly, most OOD data exits at the first exit, and some OOD data also proceed to later exits. We provide an explanation for this in the following:

- Why some ID data exit in earlier classifiers: Most ID samples exit at the *third exit*, while the majority of OOD samples exit at the *first exit classifier*. The pre-trained victim model is optimized and trained on the ID training dataset, resulting in higher prediction confidence when predicting for ID data. This enables some test ID samples to be predicted with high confidence at early exits, such as the third exit. This results in some ID data exits earlier.

- *Why some OOD data exit in later exits*: A significant portion of OOD samples exit at the first exit, providing noisy and misleading information that disrupts attacker's training. *When the generated attack OOD data are more challenging for the victim model to classify, resulting in lower prediction confidence, some OOD data may exit through later exits.* However, these harder-to-classify OOD data may differ significantly from the ID training data, and using such OOD data to extract the victim model could result in a lower-quality clone model. Additionally, a substantial portion of OOD samples still exit at the first stage, further ensuring that our method remains effective in defending against such attacks.

- *Why our method can still effectively defend when some OOD exit later classifiers*: Even for OOD samples that exit at the fourth stage, our optimization objective encourages the model to memroize the data representations by maximize the mutual information between the input data and latent representations, this worse data representations reduce the model's generalization ability to OOD data. Furthermore, we minimize the mutual information between features and labels, thus the correlation between data and labels is weakened, lowering the prediction accuracy for OOD data and achieving the desired defensive effect.

Table 12: Exit point evaluation during deployment

| Exit points | Exit0 | Exit1 | Exit2 | Exit3 | Exit4 |
|---|---|---|---|---|---|
| ID Percent (%) | 5.41 | 33.78 | 52.91 | 6.39 | 1.51 |
| OOD Percent (%) | 54.90 | 0.84 | 9.10 | 26.32 | 8.84 |

Table 13: **Victim model fine-tune time for ResNet model (seconds)**

| Method | MNIST | CIFAR10 | CIFAR100 | ImageNet-100 |
|---|---|---|---|---|
| DNF | 20.52 | 65.52 | 125.31 | 780.29 |

Table 14: **Victim model fine-tune time for ViT model (hours)**

| Method | ImageNet200 (GAN) | ImageNet200 (Indoor67) | ImageNet200 (SVHN) |
|---|---|---|---|
| DNF | 6.51 | 7.26 | 6.23 |

Table 15: Clone model accuracy on **CIFAR-10** with *ResNet34-8x* as the victim model under different loss weight.

| Weight $\tau$ | Clone model accuracy | Victim model accuracy |
|---|---|---|
| 0.8 | $58.24 \pm 1.52\%$ | $94.79 \pm 2.10\%$ |
| 1.0 | $53.91 \pm 2.30\%$ | $94.34 \pm 0.07\%$ |
| 1.2 | $56.31 \pm 1.69\%$ | $94.78 \pm 1.19\%$ |

Table 16: Clone model accuracy on **CIFAR-10** with **NAS model** as the victim model.

| Method | Clone model accuracy | Victim model accuracy |
|---|---|---|
| Undefended | $70.93 \pm 0.46\%$ | $96.58 \pm 2.01\%$ |
| DNF | $48.02 \pm 1.23\%$ | $96.52 \pm 1.95\%$ |

Table 17: Clone model accuracy under **JBDA** attack on **CIFAR-10** with *VGG* as the victim model.

| Attack | Defense | Accuracy |
|---|---|---|
| JBDA (Soft-label) | Undefended ↓ | $56.63 \pm 1.24\%$ |
| | RandP ↓ | $48.70 \pm 1.73\%$ |
| | P-poison ↓ | $42.54 \pm 2.02\%$ |
| | GRAD ↓ | $43.31 \pm 1.76\%$ |
| | MeCo ↓ | $50.02 \pm 1.28\%$ |
| | DNF ↓ | $\mathbf{40.60 \pm 1.22\%}$ |
| JBDA (Hard-label) | Undefended ↓ | $49.88 \pm 2.05\%$ |
| | RandP ↓ | $49.42 \pm 0.85\%$ |
| | P-poison ↓ | $48.51 \pm 1.97\%$ |
| | GRAD ↓ | $48.75 \pm 2.45\%$ |
| | MeCo ↓ | $48.24 \pm 2.96\%$ |
| | DNF ↓ | $\mathbf{44.20 \pm 1.24\%}$ |

Table 18: Clone model and victim model accuracy on **CIFAR-10** with *ResNet34-8x* as the victim model under different **Entropy loss weight** $\alpha$.

| Weight $\alpha$ | Clone model accuracy | Victim model accuracy (utility) |
|---|---|---|
| Without OOD | $78.90 \pm 1.23\%$ | $94.79 \pm 1.98\%$ |
| Entropy Weight = 0.1 | $52.88 \pm 3.20\%$ | $94.34 \pm 0.07\%$ |
| Entropy Weight = 0.2 | $58.71 \pm 2.02\%$ | $95.12 \pm 0.98\%$ |

## J.3 ACTIVETHIEF SAMPLING EXTRACTION DEFENSE RESULTS

Table 19: Victim and clone model accuracy with and without ActiveThief sampling.

| Methods | Teacher | Student |
|---|---|---|
| Random sampling (no defense) | $81.18 \pm 1.21\%$ | $55.49 \pm 1.50\%$ |
| Random sampling (DNF, ours) | $80.13 \pm 0.86\%$ | $15.32 \pm 1.21\%$ |
| ActiveThief uncertainty-sampling (no defense) | $81.18 \pm 1.21\%$ | $57.85 \pm 0.37\%$ |
| ActiveThief uncertainty-sampling (DNF, ours) | $80.13 \pm 0.86\%$ | $17.01 \pm 2.18\%$ |

## J.4 NOTATION TABLE AND ALGORITHM

Table 20: Table of symbol notations.

| Symbol | Description |
|---|---|
| $\boldsymbol{x}$ | Input data |
| $\boldsymbol{y}$ | Data's label |
| $l$ | Loss function |
| $V$ | Victim model |
| $C$ | Clone model |
| $D_{tr}$ | The training dataset |
| $M$ | The number of classes in the dataset |
| $\{V_i^*\}_{i=0}^{i=N}$ | The exit classifiers |
| $G_{ood}$ | Pre-trained OOD dataset generator |
| $V^*$ | The Early-Exit neural network |
| $j^{th}$ | The $j^{th}$ class probability |
| $H$ | The entropy regularization |
| $K$ | The piecewise function |
| $J$ | The piecewise function |
| $I$ | The mutual information |
| $Z_i$ | The latent features |
| $\epsilon$ | Gaussian random variable |
| $\alpha$ | Entropy loss weight |
| $\boldsymbol{\delta}$ | Network weights |
| $\alpha_i$ | The hyperparameter for the $i^{th}$ exit |
| $\beta(Z)$ | A variational approximation |
| $\mathbf{r} = \{r_0, r_1, .., r_N\}$ | Confidence thresholds |

As shown in Figure 3, We show the experiment results about the accuracy of ID data at each exit. We use 10,000 test samples and force predictions to occur at a single exit, reporting the accuracy of the early-exit network. Since OOD data lack true labels, their accuracy cannot be computed.

As shown in Figure 4, it is a diagram of a neural network with an early exit mechanism, where multiple intermediate classification exits (Exit1 to Exit4) are placed at different feature extraction stages, allowing the model to exit early when certain conditions are met to reduce computation, while retaining the main exit (Exit5) for final classification.

The figure 5 illustrates the relationship between different probability distributions and their corresponding entropy values. The distributions range from a uniform distribution with maximum entropy (2.3026) to a deterministic distribution with zero entropy. Each bar chart represents the probability associated with ten classes, highlighting how entropy decreases as the distribution becomes more concentrated on specific classes. These visualizations demonstrate the fundamental principle that entropy reflects the level of uncertainty or randomness in a probability distribution.

**Algorithm 1** DNF **Training**.

1: **Require:** Pre-trained victim model $V$ with parameters $\boldsymbol{\delta}_V$; $M$ is the number of training iterations; the in-distribution (ID) training dataset, $\mathcal{D}_{tr}$; pre-trained OOD dataset generator.
2: **for** $k = 1$ to $M$ **do**
3:     randomly sample a new mini-batch data $(\boldsymbol{x}, y)$ from $\mathcal{D}_{id}$
4:     randomly sample a mini-batch of OOD data from the pre-trained OOD dataset generator.
5:     calculate empirical IB loss for ID data (Eq. (6)).
6:     calculate empirical IB loss for OOD data (Eq. (7))
7:     calculate the overall DNF loss function (Eq. (8))
8:     update the exit classifiers $\{V_i^*\}_{i=0}^{i=N}$ of Early-Exit model $V^*$ by gradient decent with Eq. (8)
9: **end for**

**Algorithm 2** DNF **Testing**.

1: **Require:** EENN $V^*$; $\mathbf{x}$ is the input data; the array of confidence thresholds $\mathbf{r} = \{r_0, r_1, .., r_N\}$.
2: **for** $i = 0$ to $N$ **do**
3:     $V_{i,\max}^* = \max_j V_{i,j}^*(\mathbf{x}, \epsilon)$, where $\epsilon \sim \mathcal{N}(0, I)$
4:     **if** $V_{i,\max}^* \geq r_i$ **then**
5:         The inference stops.
6:         **Return** $V_i^*(\mathbf{x}, \epsilon)$
7:     **else**
8:         The inference process continues.
9:     **end if**
10: **end for**

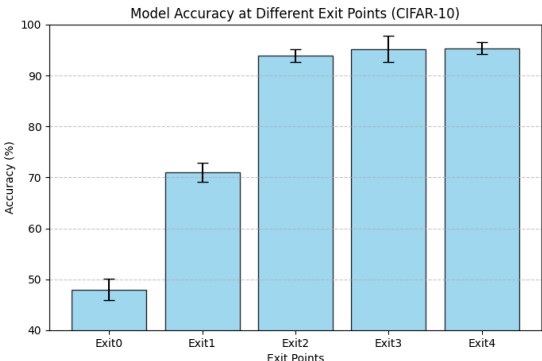

Figure 3: Model accuracy on the CIFAR-10 dataset at different exit points, utilizing ResNet34-8x as the backbone.

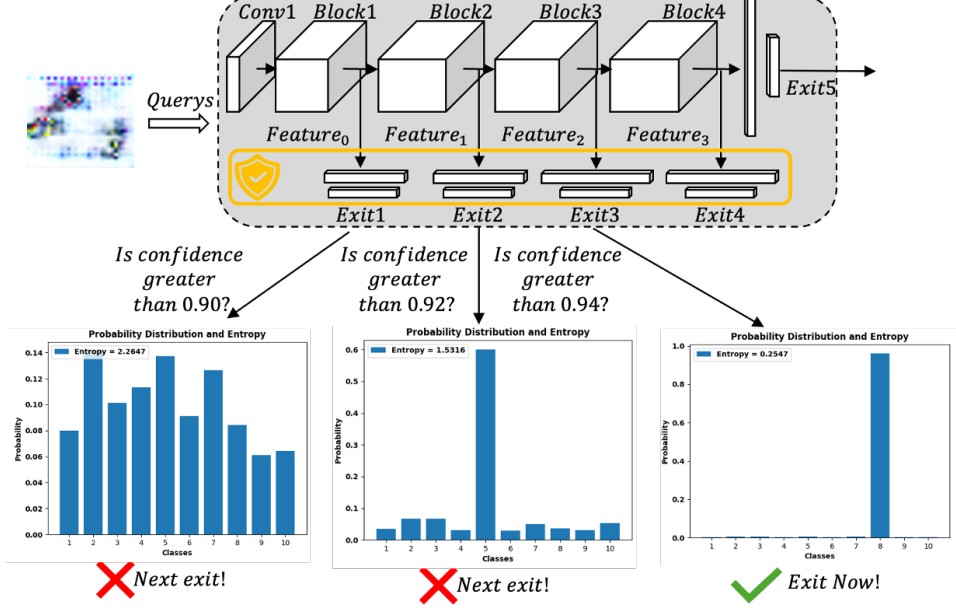

Figure 4: Neural Network with Early Exit Mechanism.

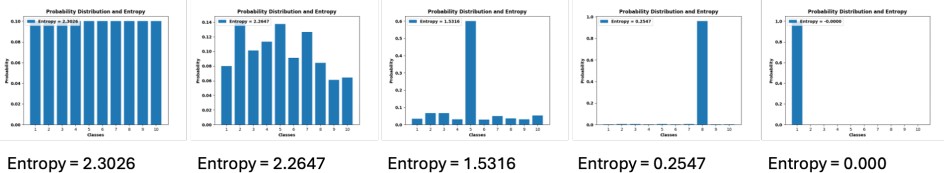

Entropy = 2.3026        Entropy = 2.2647        Entropy = 1.5316        Entropy = 0.2547        Entropy = 0.000

Figure 5: Visualization of Probability Distributions and Corresponding Entropy Values.

