# OpenReview forum: "Dynamic Neural Fortresses: An Adaptive Shield for Model Extraction Defense"
_ICLR.cc/2025/Conference — ICLR 2025 Poster_

### Official Review · Reviewer_CXR5 · 2024-10-30

**Soundness:** 2
**Presentation:** 3
**Contribution:** 4
**Rating:** 8
**Confidence:** 4

**Summary:**

In this paper, a defense against model stealing attacks (targeting either the model architecture or its functionality) based on a multi-exit neural network is proposed. The main idea is to output accurate prediction scores for ID data from the later network exits, as well as uninformative scores for OOD data from the earlier exits. To do so, for each network exit, a thresholded classifier is trained on the respective intermediate layer representation with a specifically designed loss, which maximizes the aforementioned objective using concepts from information theory. During the deployment, an exit is chosen for a sample when the maximum score of an exit classifier exceeds the respective threshold.

**Strengths:**

- The paper presents a clearly novel idea to address a very relevant issue. Indeed, to the best of my knowledge, this is the first application of a multi-exit neural network to defend against model extraction attacks.
- The proposed network architecture can also reduce the inference time during deployment.
- The approach is very intuitive and well-justified.
- The reported results are promising.

**Weaknesses:**

- 90% of IID samples exit in the first 3 exits. Although this can be viewed as a benefit (it reduces the inference time), on the other side, the defense mechanism will produce less informative outputs for those samples. The impacts of these effects should be clearly understood.
- I appreciate the fact that the authors consider different types of attacks and try to implement adaptive ones. However, a best practice when dealing with security is to simulate a worst-case scenario against the strongest attack. This helps understand the limitations of the defense and estimate lower bounds of robustness in these settings - even if, in practice, they are unlikely to occur. In this case, the adaptive attacks should be implemented using model extraction techniques that rely on some knowledge about the training data distribution. This assumption is not too unrealistic, as it might happen that the attacker (who knows the domain on which the model is applied) is able to gather in-distribution data from public domains - for instance, if the model is a malware detector, it should be very easy to collect samples and also very likely to have some overlap between them and the training data used by the victim. In other cases, the attacker might possess a subset of or all the training data, and she could easily train its own model, but she is rather interested in reproducing the exact model functionality and reproducing its decision boundaries to build a surrogate model and use it for other attacks (like evasion ones, aka adversarial examples).

**Questions:**

- Could you please estimate the impact of early exiting for IID samples? For instance, you might compute the misalignment in model outputs for IID samples when they exit early with respect to being forwarded into the entire network.
- Could you please evaluate the defense against a worst-case attacker, enhancing the already implemented adaptive attacks with (partial) knowledge of the training data distribution?

---

### Official Review · Reviewer_FTna · 2024-11-04

**Soundness:** 3
**Presentation:** 2
**Contribution:** 3
**Rating:** 6
**Confidence:** 3

**Summary:**

Model extraction is a type of attack where an attacker tries to replicate a victim model to either:
1. Estimate the model’s parameters to emulate the model’s performance
2. Copy the model’s architecture, to recreate the model as-is.
3. Get protected knowledge of the training data of the victim model, to better understand the data distribution it was trained on, so that other type of adversarial attacks can be done.

Existing defense strategies are costly – they do not differentiate between benign and malicious queries from an attacker and this form of defense allocates the same computational power to both. This paper provides a novel way to tackle model extraction attacks – Dynamic Neural Fortresses.

They propose an early-exit strategy wherein the victim model has built-in early exits routes that the model can take and provide outputs that are OOD from it’s expected input-output combination. If an input query matches an early-exits threshold, the model inference exits with the output at that stage.

**Strengths:**

1. The paper presents an interesting defenseive method to counter model extraction attacks. The paper’s novelty lies in the core idea of using a dynamic exit strategy based on the input query. While early exit strategies have been explored in the context of neural networks, their application to defensive methods is novel.
2. The paper is well written, and the core idea is simple to understand. The language is lucid but see weakness 2, 3.
3. The paper is well organized with a clear progression between sections. Figure 1 greatly aids clarity in trying to understand the pipeline, however see weakness 2.
4. Experimental evaluation is robust and does seem to support the author’s claims that DNF achieve substantial reduction is successful model cloning.
5. This paper addresses a growing concern in the space of AI/ML model deployment – protecting against model cloning and privacy and intellectual rights protection. This work does have the potential to help drive forward work in defense space for these attack types.

**Weaknesses:**

1. Despite strength 5, this method can be adapted widely only after these weaknesses are addressed and questions explored.
2. Should make better use to visual elements – probably, atleast in the appendix, add an example of what an attack query would look like, why the victim system would classify the query as attack and what the victim model’s behaviour would be on it, how early would it exit?
3. Math is useful and helps to aid the reader’s understanding but at times also hampers readability. Especially in textual sections it breaks the flow of readers. Something that may help is condense the math and limit them to equations that can be repeatedly referenced or have a table of symbol notations that readers can refer to.
4. Some sections could use with clearer explanations - OOD Data Learning Objective, underlying theory for Entropy and IB regularization. Maybe providing examples around mutual information or ER could help.
5. The paper does provide some explanation about Entropy and IB regularization but could expand a little more on how mutual information reduction leads to lower predictability and can be leveraged  for distinguishing between benign and malignant queries.
6. Maybe a comparison with other information-theory based approaches such as standard adversarial training would help drive home the imminent advantages on DNF. Another set of comparisons that could  strengthen the paper’s results are against other dynamic architectures (example ‘BranchyNet’).
7. The paper uses ER to determine optimal exits from the model’s inference. However the choice of thresholds is only briefly discussed. Maybe an ablation study of various hyperparameters, exit thresholds and entropy weights could help explain the choice a certain threshold or explain the assumptions that the authors may have made.

**Questions:**

1. Concepts related to entropy and IB regularization are presented with some mathematical rigor and learning objectives for both ID and OOD data are presented with entropy and IB regularization contratints; However some additional insights into potential limitations are necessary – How would the strategy perform under adaptive attacks with a much varied and increasingly sophisticated OOD spectrum? And how it would impact models that aim for domain generalizability and to incorporate that OOD spectrum into their model’s capabilities?
2. How does this defensive method translate to multi-modal architectures like VLMs. Or multi-pipeline methods where each branch operates on different modalities? Or ML methods where different models are trained for different modalities and their outputs are combined (via some aggregation)?

---

### Official Review · Reviewer_zh4c · 2024-11-04

**Soundness:** 3
**Presentation:** 2
**Contribution:** 2
**Rating:** 6
**Confidence:** 3

**Summary:**

The dynamic neural fortress (DNF) defense method introduced in this paper employs a dynamic early exit neural network to defend model extraction attacks. This approach effectively provides simultaneous protection for model functionality, network architecture, and enhanced defense efficiency against these threats. Extensive experiments demonstrate that the proposed defense method outperforms SOTA model extraction defenses in terms of both effectiveness and efficiency.

**Strengths:**

* The first defense framework simultaneously offers three key protective benefits: protecting the functionality, and model architecture, while improving the efficiency of the inference.

* An innovative design of the loss function is achieved by incorporating the Information Bottleneck (IB) theory.

* The experimental design is well-structured and covers various scenarios, effectively validating the method's effectiveness.

**Weaknesses:**

* The claims regarding the protection of model architecture are overstated. Early Exit (EE) mechanisms indeed prevent attackers from executing the entire pipeline of DNN, therefore protecting the entire model architecture information from being leaked. However, the authors fail to provide how attackers might exploit this vulnerability to steal the model architecture when executing the entire network. Furthermore, EE mechanisms typically occur in the last few layers of DNNs; therefore, while the proposed approach may protect certain layers, it only works those that are unexecuted, leaving the majority of the neural network vulnerable (if there are effective attacks that can steal the model architecture). The authors should consider discussing these limitations in a dedicated section titled "Limitations."

* The definitions of out-of-distribution (OOD) and in-distribution (ID) data lack clarity. It is unclear why the authors consider OOD data to be "illegal" while ID data is deemed "legal," and the rationale behind the corresponding loss term needs further explanation. Additionally, the authors aim to minimize the mutual information between $X_{id}$ and $Z_{id}$ in Eq. (3). However, this approach could potentially compromise the overall performance of deep neural networks (DNNs). The authors should provide additional clarification on why a reduced mutual information between $X_{id}$ and $Z_{id}$ does not impact the prediction accuracy.

* Table 12 indicates that queries drawn from ID dataset exit at Exit2 over 90%, while the OOD queries only exit at about 75% at the same stage. This discrepancy seems inconsistent with the motivation behind two loss terms in Eq. (3) and Eq. (4). The authors should explain this discrepancy and discuss how it impacts the effectiveness of the proposed defense mechanism. I would like to suggest the authors provide a more detailed analysis of the exit patterns for ID vs OOD data.

* The explanation for choosing a specific mutual information optimization method to achieve the defense objectives lacks a deeper theoretical explanation and intuitive justification, making it challenging to fully follow the principles behind the proposed method.

* The experiments conducted to protect the model architecture appear limited, which does not sufficiently demonstrate the contribution related to model architecture protection mentioned in the paper. Consider adding additional experiments and evaluation metrics specifically designed to assess the robustness of the model architecture against potential theft.

* It would be advantageous to include experiments that investigate the correlation between accuracy and exit points, providing a clearer visualization of the early exit mechanism's impact. I would like to suggest a graph showing accuracy vs. exit points for both ID and OOD data or report a statistical analysis of this relationship.

* It seems that all datasets utilized are classification datasets, which makes it difficult to validate the effectiveness of the proposed method in other tasks and domains.

* The notations in this article have been used repetitively, e.g., $r$.

**Questions:**

* Can the proposed defense be easily extended to other tasks and domains, such as object detection and NLP applications?

* Does the number of exit points impact the performance of the proposed defense?

* According to the design, earlier blocks are intended to reduce the model's predictive capability. However, it is notable that the ID dataset maintains high accuracy even after exiting at Exit2. This raises questions about the effectiveness of the defense mechanism. Moreover, the OOD dataset still retains 35% of its data after passing through the last two blocks. What is the observed defense effect in this case?

---

### Official Review · Reviewer_xham · 2024-11-04

**Soundness:** 3
**Presentation:** 3
**Contribution:** 3
**Rating:** 6
**Confidence:** 3

**Summary:**

This paper introduces a new defense against model extraction attack for model architecture and model utility. The key idea is to use multi-exit neural network architecture and its random exit mechanism to protect the network's architecture while ensuring the efficiency. For benign queries, the authors trains the early-exit model to distinguish OOD data (attack queries) and in-distribution data to ensure the model utility.
Finally, the authors show that DNF outperforms previous defenses and evaluate the adaptive attacks.

**Strengths:**

+ Good motivation. The authors adopt multi-exit architecture to defend architecture extraction attack, which is a well motivated and interesting idea.
+ Extensive evaluation. The authors not only evaluate the defense effectiveness but also adaptive attacks.

**Weaknesses:**

- The assumption of attack data are OOD data, although widely adopted in prior work, should be more carefully justified. Meanwhile, as the model's training data are unknown to the user, benign queries may also be OOD data. DNF might decrease the model utility in this case.
- The main part of paper (Section 4) is somehow hard to follow. I would suggest the author to simplify the notations or subscripts. Moreover, I also suggest the authors to provide an overview figure to replace some descriptions.
- Although the authors investigate the adaptive attacks, the adversary can still design more powerful attack by exploiting the multi-exit model.  Please discuss more about the potential vulnerability of multi-exit architecture and compare with prior attacks on multi-exit networks.

[1] Auditing Membership Leakages of Multi-Exit Networks. ACM CCS 2022.

[2] Model Stealing Attack against Multi-Exit Networks. arXiv:2305.13584.

[3] Mind your heart: Stealthy backdoor attack on dynamic deep neural network in edge computing. IEEE INFOCOM 2023.

[4] Aegis: Mitigating Targeted Bit-flip Attacks against Deep Neural Networks. Usenix Security 2023.

[5] Prediction Privacy in Distributed Multi-Exit Neural Networks: Vulnerabilities and Solutions. ACM CCS 2023.

**Questions:**

Can you provide a formal definition or description of in-distribution and out-distribution data in this paper's setting? How to distinguish the normal user data (OOD) and attack data (OOD)?

---

### Official Review · Reviewer_dSR3 · 2024-11-04

**Soundness:** 2
**Presentation:** 2
**Contribution:** 2
**Rating:** 8
**Confidence:** 5

**Summary:**

The paper presents “Dynamic Neural Fortress” or DNF framework as a defense against Model Extraction Attacks. These attacks allow an adversary to create a copy of a pre-trained model accessible via black-box APIs, posing risks to proprietary models. The authors identify two main challenges in current defenses: (1) Neural Network architecture protection, a thing that is taken for granted in previously proposed attacks by using the same model architecture for victim and clone models, and (2) optimizing computational resources by avoiding allocation of equal resources to both benign and attack queries.

The authors implement an Early-Exit neural network wrapper (EENN) on top of a trained model. This wrapper facilitates random exits at earlier layers for attack queries while preserving model utility by making benign queries exit at later layers. The authors assume the usage of out-of-distribution (OOD) data by attackers in most cases, but there are some experiments conducted for in-distribution (ID) data as well. Using concepts from deep information bottleneck theory, the authors optimize mutual information between input data, latent features, and output labels for training the EENN model.

The proposed method has been evaluated via testing on various architectures and datasets, and compared against other state of the art defenses.

**Strengths:**

- The proposed idea of implementing early exits as a defense against model extraction is novel and sound.
- The method is easily adaptable to different architectures like ResNets and ViTs.
- The use of entropy and information bottleneck theory is sound and well-suited to the goal of reducing extractable information for the attacker.
- The experiments conducted cover various scenarios, models and datasets validating its generalizability. The performance comparisons with state-of-the-art defenses further strengthen its credibility.
- The ablation study is thorough and captures various scenarios that highlight the effectiveness of the proposed method and its components.

**Weaknesses:**

The paper presents a technically sound idea, but the presentation is poor and needs major revisions. I am listing the weaknesses sectionwise.
### Related work:
- The related work is not organized properly, and some works are not cited in their appropriate sections, although they are cited later in the paper. For example, ActiveThief by Pal et al. (2020) [1] should be present under functionality stealing.
- When a model extraction attack is data-based, the data might be natural or synthetic. For E.g., I can generate a dataset of 10,000 images from a pretrained generative network and use that for model extraction. This would still fall under the category of data-based model extraction. Data-free model extraction means that the data used for stealing is generated based on some information received from the victim.
- Therefore, restructuring the related work section is necessary here.

### Methodology:
- The steps followed to convert a pre-trained victim model into an EENN are not easily followed. A network is trained on the ID data first. Then exit classifiers are added on top of it. Then, an OOD generator is used to generate OOD data, which is then passed through the original network without the exit networks for inference. The steps followed after this are not written in a coherent manner. One has to go through Algorithm 1 to get a clear picture of the training process.
- Overuse of the term specific to start two consecutive paragraphs (224-235 and 236-241) and even inside the paragraphs when the sentences contained in both paragraphs are not specific at all.

### Experimentation:
- The authors should differentiate between the DFME and DBME settings in more detail. In line 387, it is assumed that the reader will know that they are talking about the DFME setting instead of the soft-label setting. This also invites confusion regarding the budget difference between the soft and hard label settings, where the budget should be the same for valid comparison.
- For the DFME setting, one clone model architecture should be the same as the victim model for valid comparison (Resnet-34 in this case). Previous methods, like the prediction poisoning [2] method used by authors for comparison, have conducted experiments that keep the victim architecture for the stolen model. Moreover, the proposed method is not better than MeCo for the CIFAR-10 dataset. This should be analyzed and discussed.
- For the DBME setting, using the random strategy for sampling images is not ideal. It has been shown in the ActiveThief [1] paper that using an uncertainty-based sampling method is more effective.
- To showcase the effectiveness of the in-distribution defense, using JBDA as the attack strategy is fairly obsolete, and the paper cited needs to be corrected. The paper that proposed the attack is  [3]. The authors should use either ActiveThief or Knockoff nets attack for evaluation as they are more recent and utilize intelligent sampling-based strategies for attack. If an actual attacker has access to in-distribution data, they will try to use the best strategy possible.
- To demonstrate the defense’s effectiveness against model architecture stealing, the authors pick the latest attack by Carlini et al. but fail to show effectiveness against previously cited work, specifically “Towards reverse-engineering black-box neural networks. In International Conference on Learning Representations, 2018.” that perform attack on imagenet models. Considering that this was one of the major claims made by the authors, they should evaluate this aspect thoroughly.


### Grammar:
The paper has incoherent paragraphs, spelling mistakes, and redundant sentences. Some of them are listed below:
- Line 225, it should be “convert” instead of “covert.”
- In Table 1 and Table 2, the spelling of label is incorrect.
- Appendix D, Lines 778-779, same line repeated twice.

Citations:
- [1] Pal, Soham, et al. “Activethief: Model extraction using active learning and unannotated public data.” Proceedings of the AAAI Conference on Artificial Intelligence. Vol. 34. No. 01. 2020.
- [2] Orekondy, Tribhuvanesh, Bernt Schiele, and Mario Fritz. “Prediction poisoning: Towards defenses against dnn model stealing attacks.” arXiv preprint arXiv:1906.10908 (2019).
- [3] Papernot, Nicolas, et al. “Practical black-box attacks against machine learning.” Proceedings of the 2017 ACM on Asia conference on computer and communications security. 2017.

**Questions:**

- The authors claim their approach falls under the model extraction prevention defense category. Still, it works like a detection approach where the OOD detector is built into the model itself and thus relies heavily on the OOD data used for classification. The results shared by authors, to argue otherwise, are insufficient. I would ask the authors to include more experiments for this argument.
- If the model is trained to early exit in the case of OOD samples, but the labels used are from the original neural network (essentially the last possible exit), what is the accuracy of the model on OOD data used for training the model? I suspect that the early exit model misclassifies OOD data with high confidence. If it were learning the original network’s output labels for OOD data, then the defense would not work for the hard-label setting as the attacker would still receive a large portion of the original network’s labels as output with some erroneous ones.
- Regarding the exit point evaluation ablation study, I would like to know the accuracy at each exit and the exact number of ID and OOD samples passing through each exit instead of terms such as “over half,” etc.

---

### Meta-Review · Area_Chair_jDrQ · 2024-12-21

**Metareview:**

This paper adopted a dynamic early exit neural network to defend model extraction attacks to not only preserve the model performance but also increase inference speed. After rebuttal all reviews are clearly positive and all questions are well addressed. It should be a clear accept.

**Additional Comments On Reviewer Discussion:**

I think the authors did a good job for rebuttal, after discussion, there is no very serious issues remaining unsolved and all reviewers acknowledge this. The only interesting point is that there is one reviewer gave a score of 1 in the beginning and did not reply to rebuttal until two days before rebuttal ends. I sent an email to him to make sure he read all rebuttals and the second day he suddenly modified his score to 8 without any further questions. This is confusing to me and is too abnormal, so I decide to eliminate this review as an outlier. But other reviews and rebuttals are enough to decide. It's still a clear accept.

---

### Decision · Program_Chairs · 2025-01-22

Accept (Poster)